# Genetic Redundancy in Rye Shows in a Variety of Ways [note 1]

**DOI:** 10.3390/plants12020282

**Published:** 2023-01-07

**Authors:** Alexander V. Vershinin, Evgeny A. Elisafenko, Elena V. Evtushenko

**Affiliations:** 1Institute of Molecular and Cellular Biology, SB RAS, Acad. Lavrentiev Ave. 8/2, 630090 Novosibirsk, Russia; 2Institute of Cytology and Genetics, SB RAS, Acad. Lavrentiev Ave. 10, 630090 Novosibirsk, Russia

**Keywords:** C-value paradox, junk DNA, tandem repeats, transposable elements, subtelomeric heterochromatin, *Secale cereale*

## Abstract

Fifty years ago Susumu Ohno formulated the famous C-value paradox, which states that there is no correlation between the physical sizes of the genome, i.e., the amount of DNA, and the complexity of the organism, and highlighted the problem of genome redundancy. DNA that does not have a positive effect on the fitness of organisms has been characterized as “junk or selfish DNA”. The controversial concept of junk DNA remains viable. Rye is a convenient subject for yet another test of the correctness and scientific significance of this concept. The genome of cultivated rye, *Secale cereale* L., is considered one of the largest among species of the tribe Triticeae and thus it tops the average angiosperm genome and the genomes of its closest evolutionary neighbors, such as species of barley, *Hordeum* (by approximately 30–35%), and diploid wheat species, *Triticum* (approximately 25%). The review provides an analysis of the structural organization of various regions of rye chromosomes with a description of the molecular mechanisms contributing to their size increase during evolution and the classes of DNA sequences involved in these processes. The history of the development of the concept of eukaryotic genome redundancy is traced and the current state of this problem is discussed.

## 1. Introduction: The Concept of Genetic Redundancy (“Selfish DNA”) in Eukaryotes

Fifty years ago, in 1972, Susumu Ohno [1] formulated the well-known C-value paradox stating that there is no correlation between the physical size of an organism’s genome, i.e., the amount of DNA, and the organism’s complexity. The vision that the entire genome is functional and that the human genome should be the largest (humans are the most complex beings, are they not?) turned out to be an illusion. According to the new idea, not all DNA is important for the function and survival of an organism, and these “freeloaders” constitute rather large portions of the genomes of a very large number of species. Ohno gave it the name of junk DNA. Analyses of DNA reassociation kinetics in eukaryotic species have shown that the unique fraction of DNA consisting mainly of coding sequences makes up a very low percentage of the total DNA, while the biggest proportion of it is represented by repeated DNA sequences occurring in different copy numbers [2]. These experiments initiated the efforts to characterize junk DNA. Based on these genome size measurements involving a large number of species across diverse taxa and then revealing tremendously high variation, Ohno became confident in observing the phenomenon that he called the C-value paradox. For example, plants show 12,500-fold variation in genome size [3]; the *Drosophila melanogaster* genome is 1.2 × 10^5^ kb in size, while the human genome is 3.3 × 10^9^ kb in size, with there being little difference in the number of coding genes between them: 1.5 × 10^4^ in the former and 2.0–2.3 × 10^4^ in the latter [4]. There are many similar examples.

Further insights into junk DNA were expressed by Doolittle and Sapienza [5] and by Orgel and Crick [6], who claimed that the action of natural selection on genomes would inevitably lead to the emergence of DNA sequences that have no effect on organismal phenotype and that only have the “function” of surviving in the genome. This kind of DNA was later called, more diplomatically and accurately, selfish DNA, and Doolittle and Sapienza [5] proposed that many of them are transposable elements. The term *junk* DNA strongly implies a lack of a selectively beneficial function. The less derogatory and more to-the-point term *selfish* DNA is often used instead. Since the purpose of this paper is to emphasize that the issue is not settled, we default to the original name of *junk* DNA here. 

In the subsequent discussions on the composition of selfish DNA as well as on its emergence and preservation in eukaryotic genomes, special attention has been given to the question as to what people mean when they say “a function” in relation to DNA. Optimistic supporters of the functionality of all genomic DNA have proposed that transcribable RNAs make up a huge interconnected regulatory network possessing huge evolutionary potential, while the genome is a continuum made up of genes alternating with *cis-* and *trans-*regulatory regions [7]. It has been long established that many kinds of RNA (tRNA, rRNA) are transcribed from DNA but encode no proteins. ENCODE estimates that no less than 76% of the human genome is transcribed, with only ~1.2% of the resulting RNA being involved in encoding proteins, providing strong support to the idea that the genome is functional in its entirety [8,9]. Functions have been identified for the recently discovered other non-coding RNAs (ncRNAs), such as short miRNAs, small interfering RNAs (siRNAs) and piwi-interacting RNAs (piRNAs) [10]. However, the most abundant class of ncRNAs is the one of long non-coding RNAs (lncRNAs), which vary in length, copy number and structure, and that is why their function is not so easy to name. It has been proposed that because most lncRNAs are derived from transposable elements, both lncRNAs and transposable elements should act together in evolution [4].

A wide spectrum of assumptions and opinions as to what a “proper function” of DNA is, and, accordingly, what DNA should be considered “functional” and what should be considered “junk”, and what role natural selection has in the emergence and survival of functional and junk DNA, is typical of the ongoing dispute around selfish DNA. The considerations normally uttered rely on a limited number of facts and apply to a limited number of species. This looks natural, as there are far too many difficult-to-explain cases of apparently redundant DNA occurring in quite diverse organisms. Doolittle and Brunet [11] attempted to formulate some general criteria for identifying functional DNA. They proposed that a genome region or a DNA sequence is functional if (1) it is expressed and the phenotypic effect of this expression is detectable at the biochemical, developmental or behavioral level; (2) such expression improves organismal fitness and (3) this DNA sequence is present in past generations due to this effect of expression. The proposed criteria make the definition of “functional DNA” dependable on the selected effect (SE) [11]. Because rarely it is possible to quickly identify the function of a new DNA sequence experimentally, let alone to unravel its evolution, only a small part of large eukaryotic genomes can be seen as necessary for species survival and well-being. On the other hand, with an avalanche of incoming data about a plethora of processes—protein synthesis being one of them—which may involve DNA and transcribed RNA, such as the binding of transcription factors, DNA looping, chromatin composition, nuclear localization, etc., the meaning of the term “functional DNA” is becoming more and more elusive.

## 2. Is Subtelomeric Heterochromatin in Rye a Harbor of Selfish DNA?

### 2.1. Tandem Repeats

In the three most widespread, well-studied and commercially valuable Triticeae species (wheat, rye and barley), rye is deemed to be the most promising for the analysis of and then for discussing the concept of genetic redundancy and considering the presence of selfish DNA in the genomes. The haploid genome of cultivated rye, *Secale cereale* L., (2n = 2x = 14) is about 8.0 Gbp/1C in size [12], being larger than the genome of an average angiosperm (5.6 Gbp) [13]. For comparison, the genomes of rye’s closest relatives, barley and wheat, the haploid chromosome number in them being the same as in any Triticeae species, 7, are 5.1–5.3 Gbp and 5.8–6.1 Gbp in size, respectively. Available whole genome assemblies with annotated genes of diploid species (*Triticum urartu* and *Aegilops tauschii*, ancestors of the cultivated wheat, *Triticum aestivum*), barley, *Hordeum vulgare*, and rye showed very close coding gene counts, about 40,000 [14,15,16,17,18]. Within the genus *Secale*, there is about a 15-percent variation in total genome size between the cultivated rye *S. cereale* (the largest) and the most ancient wild rye *S. silvestre* (the smallest) [12]. The differences in size between their respective genomes correlate with the differences in the sizes of the heterochromatic regions at the chromosome ends [19]. The facts as listed above obviously suggest that the cultivated rye’s superiority in genome size amongst its closest relatives is due to a higher abundance of various classes of repeated DNA sequences. This assumption is supported by the results of the first cereal DNA reassociation kinetic experiments, which showed that more than 90% of the rye genome consists of repetitive DNA [20].

Large heterochromatic blocks in subtelomeric regions represent a chromosomal feature of rye that its most closely related genera, wheat and barley, do not possess. As recently as at the end of the past century, these regions were analyzed for DNA composition. Some tandemly organized families were found to have extremely high monomer copy numbers as was indicated by strong in situ hybridization signals. All together, these families make up 8–12% of the entire rye genome [21,22]. The molecular structure, copy number and monomer lengths were furthermore determined for the three most abundant of them, pSc119.2, pSc200 and pSc250 [23,24]. They are made up of monomeric units 118 and 379, and are 571 bp in length, respectively, with pSc200 contributing to ~2.5% of the genome and pSc250 and pSc119.2 each contributing to ~1%. Fluorescence in situ hybridization (FISH) experiments suggest that the pSc200 and pSc250 blocks coincide close to the telomere, while some pSc119.2 copies are confined to interstitial sites. The pSc119.2 sequence is also present in some other cereals, but pSc200 and pSc250 are largely rye-specific.

As far as shorter tandem repeats (minisatellites) are concerned, the most probable mechanism promoting the emergence of two or a few successive DNA monomers is the duplication event that followed replication slippage [25]. It is possible that the same mechanism applies to longer monomers, with sizes close to those of tandem families in the rye genome. By sequencing an extended DNA region with BAC119C15 in it [26], we revealed a consistent pattern of the alteration of pSc200 monomers having an average of 93% homology, suggesting an initial duplication, the divergence of the primary sequences of the monomers within the dimer and the subsequent amplification of the dimers. It is likely that virtually any DNA sequence can serve as the initial monomer; several cases of tandem DNAs emerging from pieces of transposable elements have been described [27,28,29]. Dimer amplification and the preservation of tandem repeat arrays can take place in the course of replication or mitotic/meiotic recombination by means of multiple recombination events, such as unequal crossing over between sister chromatids, sequence conversion, translocations that exchange material between non-homologous chromosomes and transpositions [30,31,32]. FISH on meiotic chromosomes shows that each of the families, pSc200 and pSc250, is present on the arms of each chromosome as a separate domain, the size and varying stain intensity of which implies the presence of more than one array within each domain. This assumption is supported by blot-hybridized patterns after pulsed-field gel electrophoresis (PFGE) of DNA in BAC clones [26]. The maximum size of the pSc200 and pSc250 arrays is 550–600 kb [33]; however, the majority of the arrays in each rye chromosome are rather short, not longer than 80–100 kb.

The chromosomal domains consisting of pSc200 and pSc250 arrays are normally separated by non-tandem DNA sequences. Restriction analysis of DNA of the BAC clones in the BAC library of the short arm of rye chromosome 1 (1RS) followed by blot-hybridization showed that the pSc200 and pSc250 arrays, as with human alpha satellite DNA, develop HORs (higher order repeats), each consisting of two to eight monomers. A single HOR is longer than 3 kb (379 bp × 8) in pSc200 and is nearly 3.5 kb (571 bp × 6) in pSc250. The order of HORs and the ratio of different HORs are specific for each array within a single arm [26]. This implies that multiple recombination events had been taking place independently in different arrays of the same chromosome arm, leading to the emergence of HORs.

The tandem families pSc200 and pSc250 have different evolutionary histories. Some pSc200 copies have been found in hexaploid wheat, other Triticeae species [34,35] and more distantly related cereals, such as rice and oats. It gives grounds to assume that this family emerged about 45 MYA, when the rice and oat lineages split [36]. The pSc250 family is younger; its copies, very low in number, are found only in the Triticeae species [35], suggesting that its age is about 15 Myr. However, when scrutinizing the evolutionary history of the pSc200 and pSc250 families, one should remember that the amplification (expansion) of these families was not observed before the radiation of the genus *Secale* (1.7 MYA) and was running especially high in the cultivated rye (*S. cereale*) [19]. Thus, pSc200 and pSc250 did not take as long to expand as, for example, human alpha satellite DNA did, which is confirmed by high (not less than 90%) homology between the monomers within each of these families [26]. The evolutionary histories of these two families are strikingly different from that of the third high copy number family of tandem repeats, pSc119.2. This family is much more common across the tribe Triticeae than pSc200 and pSc250 are. Not only rye, but also various *Triticum* and *Aegilops* species, as well as most wild barley species [37] including *Hordeum bulbosum*, which is closely related to the cultivated barley, contain this family in a high copy number. A large number of variant monomers with different lengths and highly heterogeneous primary structures are an indication that sequence homogenization has never taken place in pSc119.2 [37]. By far the most surprising pSc119.2-related observation is that both *H. vulgare* ssp. *spontaneum*, which is an immediate ancestor of the cultivated barley (*H. vulgare*), and the cultivated barley itself are devoid of this family [38]. Thus, the beginning of the barley domestication process caused the mass deletion of thousands copies of pSc119.2 genome-wide in the predecessor species.

### 2.2. Transposable Elements

Transposable elements (TEs) are the predominant component of such large genomes as the cereal ones [39,40]. For this reason, we were not surprised to see that the sequencing of the DNA around the arrays of the pSc200 and pSc250 family tandems in BAC clones revealed various classes of these elements, basically LTR-containing retrotransposons. Are these two main classes of repeats in heterochromatic regions related in some way? Is the composition of the DNA surrounding the arrays of the tandem repeats specific to them—or is it unrelated to them? To answer these questions, we used 454 reads, which served as a basis for the rye genomic library [41]. The benefit of this approach is that no contigs are needed, which, if they were, would represent a major problem in duly profiling repeats, as the genome crawds with them. Nevertheless, the average length of the 454 reads is sufficient for spotting junction regions between tandem monomers and the DNA that surrounds them. Three sets of 454 sequencing-based reads were produced (Figure 1): two consisted of reads, which were essentially DNA in the “non-tandem DNA—pSc200 (or pSc250) tandem” junction regions, and the third sample consisted of reads with DNA residing in the rest of the genome [26]. Our interest was to find out whether there was a correspondence between the copy number of separate TE families in the “non-tandem DNA-tandem DNA” junction regions and their copy number throughout the entire rye genome. The comparison was carried out using a database of eukaryotic repetitive elements, Repbase and the Triticeae Repeat Sequence Database (TREP). Our analysis of the genome composition for rye showed that 73% of all TEs identified belonged to the *gypsy*-like superfamily of LTR-containing retrotransposons. Of all the TE families identified, the most abundant was the *gypsy*-like family *Sabrina* (Figure 2), which makes up about 15.5% of the total number of TEs identified genome-wide, which greatly exceeds the abundance of the next two top TE families, the CACTA superfamily and the *gypsy*-like family, *WHAM*. At the same time, the abundance of *Sabrina* dropped abruptly when this family was flanked by arrays of tandem repeats. Some TE families, common in the rye genome, virtually do not occur flanked by arrays of tandem repeats. For the example, neither *CACTA* nor *Cereba* nor *Derami* nor *Sumana* were flanked by pSc250, and *Fatima*was never flanked by pSc200.

Thus, a surprising feature revealed by comparing the abundance of the DNA sequences in the entire rye genome and when these sequences were flanked by the arrays of the tandem monomers pSc200 and pSc250 was a nearly reciprocal replacement of TE families. In terms of genome-wide abundance, *Sabrina*, *CACTA* and *WHAM* yield precedence to *Daniela* and *Olivia* flanked by pSc200 and *Laura* and *Gypsy*-13_TA-I flanked by pSc250, but the abundance of solo-LTRs designated as *Xalas* or *Xalax* is just rocketing, especially when these are flanked by pSc250. As the ectopic exchange model suggests [42], this should be indicative of very frequent ectopic exchanges in the immediate neighborhood of the tandem arrays pSc200 and pSc250 while these arrays were in the making. The main sources of solo-LTRs are probably unequal cross-overs and the within-chromosome ectopic recombination between the LTRs of the same or different elements, provided that they share quite extended homologous regions, such as those shared by *Xalas* and *Xalax*, and *Daniela* and *Olivia*. Vicient et al. [43] distinguish four forms of retrotransposon recombination. One of these forms is the LTR-LTR recombination, which results in solo-LTRs or tandem arrays consisting of LTRs and internal domains. According to this mechanism, if an LTR borders with any other DNA sequence, for example, pSc200 or pSc250 monomers, then the LTR-LTR recombination will generate a tandem array of these monomers next to the LTR.

While the mechanisms of recombination processes occurring within the arrays of tandem monomers and leading to the formation and genome-wide distribution of higher-order repeat units, varying in length and internal organization, have been known for decades, little is known about a potential impact that the immediate DNA neighborhood of the tandem monomer arrays might have on these processes. Our analysis of the junction regions between TEs and the monomers of the tandem arrays pSc200 and pSc250 revealed that most TE transpositions occurred either directly to the monomers of the tandem array or next to them following ligation, which added a very short spacer, DNA 1–10 bp in length—a situation typical of non-homologous end joining (NHEJ). About 90% of these junction regions put together pSc250 and two groups, *Laura* and *Xalas*, the most abundant when surrounded by TEs: about 70% for pSc200 and *Daniela*, and 58% for pSc200 and *Olivia*.

In many organisms, recombination events occur at certain sites, the so-called “hot spots”, which have a particular nucleotide context [44]. In the nucleotide context of DNA of the transposable elements bordering the tandem arrays pSc200 and pSc250, motifs that could participate in recombination other than HR (homologous recombination) or NHEJ (non-homologous end-joining) and promote the propagation of the arrays of tandem monomers were revealed to occur with a probability much higher than could have been expected by random chance [26]. The lengths of these motifs are 8-12 bp, which are sufficient for recombinases to start aligning a single-strand DNA with a homologous duplex elsewhere in the genome, which can promote recombination. Thus, the LTRs containing microhomologous DNA within this TE inserted next to the pSc200 or pSc250 monomers could, together with captured monomers, recombine these sites with other copies of the same TE occurring elsewhere in the genome and thus promote the distribution of the tandem monomer arrays.

### 2.3. Rye Genomic Libraries Detailed the Organization of Subtelomeric Heterochromatin

The rye genomic libraries generated in recent years using modern DNA sequencing techniques and contig assemblies [17,18] allow for large genomic regions to be analyzed. As is known, the heterochromatic chromosome regions and, first of all, long arrays of tandemly organized monomers are especially difficult to read. Nevertheless, the contigs in the genome libraries mentioned are long enough to shed light on the structural organization of the subtelomeric heterochromatic regions of the rye chromosomes, including both classes of repeats: monomer tandems and copies of TE families. Some contigs begin from the chromosome ends, as the presence of the contig region containing telomeric monomers (TTTAGGG)n suggests. The telomere monomers are immediately followed by the array of the pSc119.2 monomers on one arm of chromosome 3R [18] and the pSc200 array on the other [17]. However, the arrays of telomeric and subtelomeric repeats are not immediate neighbors in all chromosomes. In the rye line ‘Weining’, the chromosome 7R telomere and the array of pSc250 monomers are separated by four pieces of the transposable elements *Xalax* and *Gypsy*13_TA-I, and the pSc250 array itself is interspersed with pieces of various transposable elements. The arrangement of the arrays of all of these tandem repeat families, pSc200, pSc250 and pSc119.2, being relative to each other may be extremely varying; they may (1) lie very close to each other, (2) be separated by variously sized tracks consisting mostly of pieces of diverse TE families or (3) occur in different combinations.

The schematic in Figure 3 shows one of the longest scaffolds, s291 (line ‘Lo7’), as an example to illustrate the structural organization of a chromosome region and to list all of the above-described main characteristics of subtelomeric heterochromatin. The pSc200 cluster in it is 258.3 kb in size and consists of several long monomer arrays interrupted by three regions. One of these regions, which is about 17 kb in length and located at the beginning of the scaffold, is populated by several rearranged copies of the *copia*-like family *WIS-2*, while the other two, being rather short, contain some of the *WIS-2* sequences and some of the *gypsy*-like family *Olivia*. From position 80 kb, the array of pSc200 monomers forms an 1897-bp-long region consisting of 13 HORs (purple in the schematic), each consisting of five monomers.

It is possible that more improved sequencing techniques and the assembly of genomic libraries will allow us to reveal genes within the subtelomeric heterochromatin of the rye chromosomes. Some genes have been found and described within small regions of constitutive heterochromatin in *Drosophila melanogaster* [45]; however, these regions do not have powerful arrays of tandem repeats in them, where as such regions in rye chromosomes do. The chicken pan-genome constructed from 20 de novo assembled genomes with high sequencing depth [46] brings hope that functional genes within the subtelomeric heterochromatic regions of rye chromosomes will eventually be found. As a result, the authors found 1335 previously unannotated protein-coding genes, the majority of which were located in subtelomeric chromosome and minichromosome regions and were surrounded by huge arrays of tandem repeats that made sequencing impossible. Looking over the amount of knowledge that we currently have about the structural organization of subtelomeric heterochromatin in rye chromosomes, we must confess that it is difficult to make assumptions about and understand how the long arrays of tandem DNA repeats that are prevalent in it and occur genome-wide in thousands of copies, in which ever-running recombination processes affect the neighboring TEs, most of which appear in the form of relatively short pieces such as solo-LTRs, could participate in encoding, regulation or any other molecular process related to survival, reproduction or behavior, or any other process that is beneficial to the metabolism and well-being of living organisms. It is therefore logical and reasonable to assume that what DNA in such chromosome regions does is only ensure its survival, which is consistent with the criteria of “selfish DNA” and the concept of genetic redundancy in eukaryotic genomes.

## 3. Gene Duplications

Can we, based on our current level of understanding of the molecular processes unfolding in cells, answer the question as to whether DNA in the subtelomeric heterochromatic regions of the rye chromosomes is the only repository of the sequences, whose possible functions are not yet known? Gene duplications that occur due to polyploidization following whole genome doubling (WGD) or due to local gene duplications (small-scale duplication, SSD), are often considered to be possible sources of evolutionary novelty and adaptation [47,48]. This opinion is based on frequent cases of deletion of new copies that had not acquired new functions [48]. However, some studies show that neither acquisition of new functions (neofunctionalization) nor delegation of complementing parts of the original function to both copies (subfunctionalization) are indispensable for the survival of both copies in a genome. An analysis of 901 SSD-derived gene pairs in *Brachypodium distachyon*, *Oryza sativa* and *Sorghum bicolor* showed that only 23.8% of the resulting copies had acquired new functions, 0.4% underwent subfunctionalization and 15.2% underwent a rapid specialization followed by neofunctionalization to the effect that both copies acquired functions that were unlike each other’s and unlike their ancestral genes’ [49,50]. The highest percentage of the genes’ second copies, 60.6%, performed the same function that their originals did, i.e., supported the existing function [49].

We have not been lucky enough to learn from the literature how many duplicated genes there are in the rye genome. For this reason, we will confine ourselves to the results that we have obtained from analyzing the structure and expression of genes for the centromere-specific histone H3 variant (*CENH3* in plants). The CENH3 proteins encoded by these genes play a universal, important role in cells as they determine the position of the centromere in chromosomes. Any error in the transcription, translation, modification or transport of this protein can compromise the formation of active centromeric chromatin, leading to impairments in kinetochore assembly and cell division. We found that the rye genome, as with wheat and barley, contains two genes, each encoding a separate form of the protein, αCENH3 and βCENH3, differing (1) in size due to an extended deletion in the N-terminal domain of βCENH3 and (2) in intron–exon structure [51]. An intriguing aspect of *CENH3* evolution is that some well-studied and very common cereal species, such as rice and maize, are quite comfortable with possessing only one form of the protein and, accordingly, with possessing only one copy of the gene to encode this protein [52,53].

An analysis of genome and transcriptome libraries for 23 cereal species made it possible to establish that the evolutionary process leading to the formation of the two-copy system of encoding CENH3 was confined to the *CENH3* locus, which emerged about 50 MYA in a common ancestor of the subfamilies Bambusoideae, Oryzoideae and Pooideae [54]. An example of the initial organization of the *CENH3* locus is rice (*Oryza sativa*), in which the locus consists of the syntenic genes *CDPK2*, *CENH3* and *bZIP* and is small, at 15.3 kb (Figure 4). The *βCENH3* gene was for the first time found as a part of the locus in the species of the tribes Stipeae and Brachypodieae and the subfamily Pooideae; it emerged about 35–40 MYA. The duplication was accompanied by changes in the intron–exon structure. Figure 4 and Table 1 present the results of analysis of the structure of the *CENH3* locus in some cereal species representing the most important stages in the evolution of the subfamily Pooideae. The main trend in the evolution of the locus is its growth due to the growth of the spacer between the *αCENH3* and *βCENH3* genes, with the spacer growing in parallel with the genomes. This trend is a feature shared by the Pooideae branch leading to species in the tribe Triticeae (barley, rye, wheat) and by the branch leading to species in the subtribe of Aveninae (oats, *Avena sativa*, Figure 4). In *Bromus tectorum* (the tribe Bromeae, which shares a common direct ancestor with Triticeae [55]), IS2 is already as large as nearly 55 kb in size. At the same time, in the common direct ancestor, an inversion happens to the *βCENH3* gene. IS2 grows in size due to a mass introduction to it of various families of LTR-containing retrotransposons: *gypsy* and *copia*, and—albeit to a lesser extent—transposons of the CACTA superfamily. The TE families largely occur as short pieces of DNA, with the largest share of clusters consisting of the copies of diverse families imbedded in the copies of other families. Some copies of TEs and tracks of simple repeats without signs of substantial rearrangements were noted, too, but they were few in number. The further evolution of the *CENH3* locus within the tribe Triticeae is characterized by the preservation of the *CDPK2* gene located to the left of *βCENH3* in the subtribe Hordeinae species (barley) and by the replacement of *CDPK2* with *LHCB3-l* in the subtribe Triticinae species (rye, wheat) (Figure 4). The locus and IS2 further grow in size, reaching their respective top values of 218 kb and nearly 190 kb. The domestication process, too, leads to an increase in IS2 size, which follows from a comparison of its sizes between (1) the cultivated barley (*Hordeum vulgare*) and its immediate ancestor wild *H. vulgare*, ssp. *spontaneum*, and between (2) the genome A of the cultivated wheat (*Triticum aestivum*) and the donor of wheat’s A genome, *T. urartu*. Each intergene spacer in each species, including evolutionarily close species in the tribe Triticeae, is characterized by its specific set of TE families [54], suggesting that it is very unlikely for them to possess any evolutionarily fixed functions.

The formation of the intergene spacers in the *CENH3* locus and the process of IS2 expansion, which accelerated during Triticeae domestication, are, in a way, similar to the process of expansion of the heterochromatic regions of the rye subtelomeres. The clusters consisting of multiple degenerate copies of various TE families transposed to each other look like the monomers of tandem repeats put together in HORs. A mix of diverse domains belonging to diverse TE families, which alternate with short arrays of micro- and mini-satellite DNA as well as AT-rich regions, indicates that violent recombination events were very frequent within the locus in its past evolutionary history. Whether the effect of duplication triggered the above processes around the *CENH3* paralogs during the evolution of the genomic region contained in the *CENH3* locus remains unknown. If we looked into many more regions such as this (that is, the ones possessing these paralogous genes), we would perhaps find the key.

It would be especially exciting to analyze the expression of the paralogous genes *αCENH3* and *βCENH3*. The level of expression of both was directly correlated with the intensity of cell division at different stages of development of two phenotypically different rye cultivars, ‘Imperial’ and K69 [56]. The level of transcription of *αCENH3* was much higher than that of *βCENH3*, especially during sprouting and in the reproductive tissues, when and where division is much more intensive than it is in leaves and stems. In stems, where the intensity of cell division is the lowest, the expression of these genes falls to its minimum basal level, and there the level of transcription of *βCENH3* becomes higher than that of *αCENH3*. Next, the transcription products and the αCENH3 and βCENH3 proteins were transported to cell nuclei and were included in the nucleosomes of centromeric chromatin [56]. These results, however, are not sufficient for us to expect evidence of neofunctionalization, subfunctionalization or specialization in these paralogous genes. The pressure exerted on the coding part of the *βCENH3* gene due to purifying selection [54] led to the preservation of its conserved function, and the benefit of possessing two genes perhaps consists of the gene dosage effect in the maintenance of the balanced total number of the CENH3 proteins, which serves to maintain the necessary level of cell division intensity at various stages of plant development. Another advantage is that each paralog encodes an N-terminal tail (NTT); so, the resulting NTTs have substantial differences in amino acid sequences [51], thus maintaining their stoichiometric relationship with partners in multicomponent interactions in changing external conditions [48,57]. As Fagundes et al. opine, if a genetic element (the *βCENH3* gene in our case) supports the organism’s adaptive level, it is qualified as functional and its “supporting function” is a proper biological function [58].

## 4. Conclusions: The Concept of Genetic Redundancy in Eukaryotes Revisited

Over the decades that have elapsed since the C-value paradox and junk DNA were proclaimed, many attempts have been made to propose a role—in a manner that was well substantiated at each particular point of time—for the excess DNA—or at least for some of it. The first attempt consisted in the division of total DNA into two categories: DNA in one of them encodes proteins; DNA in the other category controls the cell volume and cell nuclear volume [59]. The hypothesis about the latter function was put forward based on the correlation between these values and the total amount of DNA in many species. However, we have yet to know what the molecular mechanism underlying the latter function is. We mention this attempt because if the existence of long arrays of tandemly organized DNA monomers had been known of at that time, it would be logical to give this class of sequences a starring role in the performance of said function. However, a correlation does not necessarily mean that one trait (a factor) defines the characteristics (the size) of another. We have searched the literature for reasonable assumptions about possible functions of the above-described long tandem DNA regions, which form part of the subtelomeric heterochromatin in rye, but to no effect. For this reason, we will next proceed to discuss other classes of DNA sequences, which may potentially be defined as junk DNA.

Until recently, transposable elements were at the center of discussions about whether they may possess a useful function as regulators of gene activity. The most frequently nominated candidates were SINEs (in particular, the *Alu* family) because of their wide occurrence in mammalian genomes and rather small sizes [60]. However, neither LINEs nor SINEs are very abundant in plants. In particular, their percentages are infinitesimally low in the large genomes of Triticeae species which are abundant with other TE classes. The other TE classes have normally been regarded as having a possible potential utility of useful genetic material, which could in the future result from DNA rearrangements and be due to novel genes that might arise in these TEs. Such assumptions have been cutely characterized as being “theological” [11], for evolution has no foresight [61]. The assumptions about the possible participation of TEs in the regulation of gene expression were limited to those DNA regions and those TEs that occurred close to coding DNA sequences [62]. Not only did these assumptions ignore tandem monomer arrays, they likewise ignored extended TE clusters measuring dozens of kilobases in size, which appear as a mess of fragmented copies belonging to diverse LTR-retrotransposon families. For example, DNA regions similar to the spacers between the *αCENH3* and *βCENH3* genes in Triticeae species are characterized by the species-specific sets of TEs without signs of conservatism or collinearity, the two properties being characteristic of the coding regions in these species.

After the publication of ENCODE results [8], the focus of discussions on genetic redundancy shifted to the type of RNA known as long non-coding RNA (lncRNA). Their characteristics, such as occurring in a larger copy number than any other transcript and being diverse in length, amount and structure, make it somewhat difficult to single out a universal function and at the same time make it easy to produce more assumptions. For example, those who advocate for the total functionality of lncRNAs see these sequences participate in a broad range of functions related to the chromosome architecture, the expression of genes in the three-dimensional structure of the nucleus, signal transduction and cell migration [63]. On the other hand, their opponents suggest that differences between functional, non-coding RNA and junk RNA be identified first, and as long as the line has not yet been drawn, the researchers should abide by the null hypothesis: “An uncharacterized non-coding RNA likely has no function, unless proven otherwise” [64]. The estimates that less than 10% of the DNA in the human genome is under purifying selection [65,66], while nearly 80% of the entire genome is subject to transcription, led to the conclusion that at least 87% of transcribable DNA produces junk [67]. This conclusion follows from the criterion proposed by Palazzo and Koonin: if an RNA molecule has a positive effect of adaptation under the pressure of purifying selection, no matter how weak the pressure, the molecule is functional [67]. The transcripts that do not meet this criterion deliver material for a large number of strongly diverse lncRNAs to evolve via the non-adaptive mechanism of neutral evolution, and so these transcripts should end up as functional lncRNAs. Considering widespread transcription events (80% of the genome is subject to transcription), almost 90% of the junk RNAs represent intermediate long non-coding RNAs. “However abhorrent the concept of junk DNA might be to many biologists, this conclusion is inescapable”, state Palazzo and Koonin [67].

With his C-value paradox, Ohno [1] disillusioned us about the functionality of the entire genome. Progress in understanding genetic redundancy makes us confess that natural selection is not so immensely powerful as it used to seem and that this power does not always work exclusively for the cause of adaptation. Progress in molecular biology provides makes Doolittle’s and Brunet’s statement [11] that “evolution by natural selection operates independently (and sometimes oppositely) at different levels of the biological hierarchy (gene, cell, organism, species)” sound more and more convincing. We believe that the word “gene” in this statement can be fairly replaced with “various classes of DNA sequences”.

## Figures and Tables

**Figure 1 plants-12-00282-f001:**
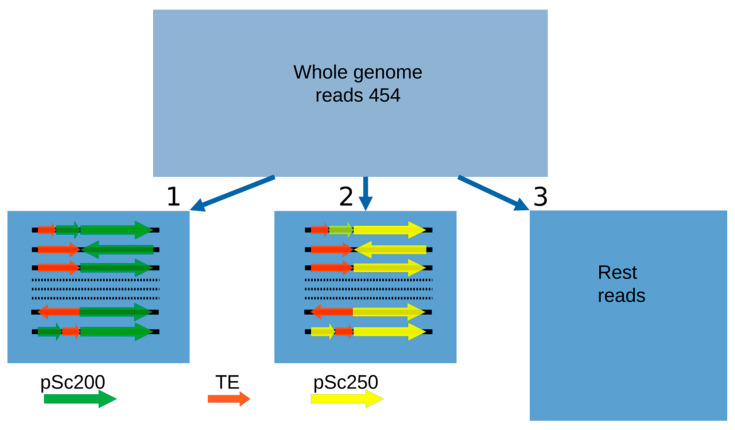
Three sets of 454 reads used for profiling TE families: (1) in “non-tandem DNA-pSc200 tandem” junction regions, (2) in the “non-tandem DNA—pSc250 tandem” junction regions and (3) in the rest of the rye genome. By way of illustration, Rectangles 1 and 2 display different possible variants of the reads.

**Figure 2 plants-12-00282-f002:**
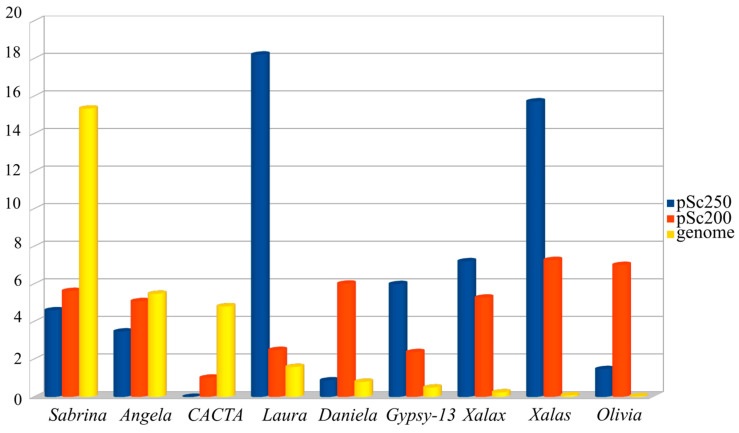
Presence of the most well-represented TE families in the rye genome (*Sabrina, Angela, CACTA*, yellow bars) and in the reads containing regions of junctions between non-tandem DNA and the monomers of pSc200 tandem repeats (*Olivia, Daniela, Xalas, Xalax,* red bars) and pSc250 tandem repeats (*Laura, Xalas, Xalax, Gypsy-13_TA-I*, blue bars).

**Figure 3 plants-12-00282-f003:**
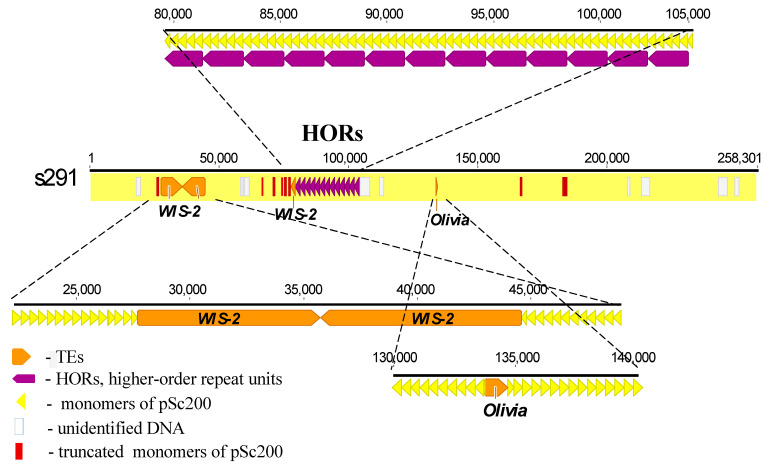
Schematic of the molecular organization of heterochromatin in rye subtelomeres (line ‘Lo7’), with the scaffold s291 as an example. pSc200 monomers (yellow finger-post arrows) and transposable elements (brown finger-post arrows) alternate with pSc200 monomers. Thirteen HOR units, each consisting of five pSc200 monomers, are pointed to by purple arrows. Truncated pSc200 monomers (red rectangles) are scattered along the scaffold with increased concentration near the 5’-ends of the HORs and the truncated copy of *WIS-2*. The numbers at the top show distances in kilobases.

**Figure 4 plants-12-00282-f004:**
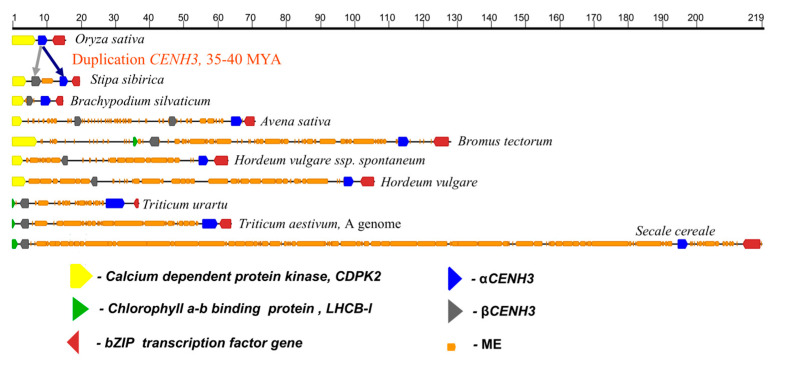
Evolution of the *CENH3* locus from its emergence in the ancestor of rye (*O. sativa*) to the domesticated species in the tribe Triticeae. The numbers at the top show distances in kilobases. The genes are calcium dependent protein kinase, *CDPK2* (yellow finger-post arrows); chlorophyll a-b binding protein, *LHCB-l* (green finger-post arrows); *bZIP* transcription factor gene (red finger-post arrows); *αCENH3* (blue finger-post arrows); *βCENH3* (gray finger-post arrows) and transposable elements (brown rectangles).

**Table 1 plants-12-00282-t001:** Size of the *CENH3* locus and its structural elements (genes and intergene spacers, ISs) in various cereal genomes *.

Species	Locus *CENH3*, kb	IS1, kb	IS1, %	IS2, kb	IS2, %	IS3, kb	IS3, %	Genes, kb	Genes, %
*Oryza sativa*	15.32	0.95	6.20			1.53	9.99	12.82	83.68
*Stipa sibirica*	19.87	1.59	8.00	5.88	29.60	1.03	5.20	11.37	57.20
*Brachypodiumsylvaticum*	14.85	0.80	5.40	2.61	17.60	1.50	10.10	9.93	66.90
*Avena sativa*	67.02	15.51	23.14	43.61	65.07	0.67	1.00	12.91	19.26
*Bromus tectorum*	127.34	32.47	25.50	69.86	54.86	7.07	5.55	17.27	13.56
*Hordeum spontaneum*	62.96	11.14	17.70	38.34	60.90	1.64	2.60	11.84	18.80
*Hordeum vulgare*	106.24	18.91	17.80	71.82	67.60	0.21	0.20	15.30	14.40
*Triticum urartu*	37.23	1.86	5.00	23.49	63.10	0.86	2.30	11.02	29.60
*Triticum aestivum,* Agenome	64.44	1.80	2.80	50.84	78.90	0.84	1.30	10.95	17.00
*Secale cereale*	218.32	0.70	0.32	189.55	86.82	16.16	7.40	11.84	5.42

* Figures are based on entries in genomic libraries within databases. *Oryza sativa*, *Stipa sibirica*, *Avena sativa*, *Hordeum spontaneum*, *Hordeum vulgare*, *Triticum urartu*, *Triticum aestivum*, A геном, *Secale cereale*—NCBI (https://www.ncbi.nlm.nih.gov, accessed on 11 October 2022), *Brachypodium sylvaticum*—Phytozome (https://phytozome.jgi.doe.gov, accessed on 14 October 2022), *Bromus tectorum*—CoGe (https://genomevolution.org, accessed on 20 October 2022). IS1 is the genome region (intergene spacer) between the left-hand marker of the locus, *CDPK2* and the *βCENH3* gene. In *T. urartu*, *T. aestivum* and *S. cereale*, the *CDPK2* gene is replaced with *LHCB3-l*. IS2 is the intergene spacer between the *βCENH3* and *αCENH3* genes. IS3 is the intergene spacer between the *αCENH3* gene and the right-hand marker of the locus, *bZIP*.

## Data Availability

Not applicable.

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
