# Peer review of "Genetic Redundancy in Rye Shows in a Variety of Ways†"

_plants, 2023, doi:10.3390/plants12020282_

Round 1
Reviewer 1 Report
The present review “Genetic redundancy in rye shows in a variety of ways” by Vershinin et al., investigated the genomic organization of 19 rye chromosomes with a description of the molecular mechanisms contributing the evolution of genomic structure. The work is well written, the results are very interesting and worthy of publication.
Minor issue: improve English language
Reviewer 2 Report
This is an excellent paper.
Vershinin and co-authors aptly describe some of the central questions and hypotheses surrounding "junk" DNA, and use a few elegant little tests on recent datasets (from the grasses---some of the best taxa to address these questions with) to motivate an insightful and timely discussion on what the field should be discussing, how to get stuck into analysing the central questions, and what the early results might be suggesting.
The paper is of utmost relevance at this time. With genomes of this quality finally revealing the intergenic landscape and allowing comparative studies thereof, this topic is going to boom.
While there is the occasional bit of clunky English in need of copy-editor attention, the style is a great mix of didactical and scientific, and the occasional injections of casual-but-unambiguous language are a truly welcome relief from the now-conventional, hyper-formal tone that (ironically) makes it so hard for authors to clearly express themselves.
The content is a good mix of review and scientific report, and the structure is a masterclass in the use of narrative in scientific writing.
I have read the draft carefully and critically, and have very little to criticise. I will offer a few minor/technical suggestions and philosophical comments.
"The preferred term in the modern literature is junk DNA" l.55
I feel this is less and less the case, I would rephrase this sentence to make the conflict explicit, e.g. "The term junk DNA strongly implies a lack of selectively-beneficial function, and is being superseded by selfish DNA, which implies a negative fitness impact at the organism level. Since the purpose of this paper is to emphasise that the issue is not settled, we default to the original name of junk DNA here"
Sentence beginning "Within the genus" l.112, "about" --> "at least".
p.5 This took me longer than necessary to get (see attached). The description needs to be workshopped.
- The questions is asks are good.
- It then needs to explain that a survey of the genome was done to test how frequently tandem repeats fell near TEs.
- Then, it needs to say why annotating 454 reads was the chosen approach. I assume this is because it is has A) A good random representation of the genome [c.f. assemblies, which collapse repeats] B) Sufficient length to spot TEs and tandem repeats within the same read, and C) Sufficient coverage to be statistically meaningful. If so, then explain that.
- Then explain the approach--an extra diagram like the ones I scribbled might really help here, it's very hard to follow as it.
- You mention "find ... correlation", which set me off to expect a scatterplot and some stats. There is no need for either--what you do is fine and makes the point well, I would just avoid the specific word "correlation" unless an r^2 value is coming up. "Correspondence", maybe?
The CENH3 and centromere arrangement bits are good, especially the discussion that makes a tentative conclusion, admits to speculation, and mentions possible confounding sources (genome assembly shortcomings) ... good science.
The only thing lacking from the concluding remarks is recognition of multi-level selection. It is true that "Evolution has no foresight" but lineage-level selection can certainly make it look like it does, and in the case of a population of TEs, each individual could be conceptualised as a group, or indeed subgroups of a population that share a common TE profile. Those individuals or sub-populations whose TE profiles play nicely with the rest of the genome in an on-average more selectively-beneficial way could lead to the proliferation of those kinds of TEs at the population level.
Here are some papers to consider in that direction:
https://sci-hub.ru/10.1016/j.tree.2011.01.006
https://www.journals.uchicago.edu/doi/epdf/10.1086/656905
Adding a few words about this will I think make the paper more rounded and (very deservedly) broaden the scope of the discussion it kicks off.
My congratulations to the authors. Given a few small tweaks as described, I unreservedly recommend this paper for publication.

Reviewer 3 Report
The manuscript aims to fulfill an important role in long-running scientific debates - revisiting old concepts from the current state of knowledge and understanding of biological processes and functions underlying them. More specifically it deals with the so called C-value paradox, i.e. the lack of correspondence between the amount of DNA, and the complexity of the organism. While appreciating the idea of bringing this to the attention of the reader, I have several remarks and comments on the approach taken and the depth of discussion performed.
The authors seem to be of the view that the current consensus is that the large sections of the non-coding DNA are defined as "junk DNA" (Lines 55-56). However, multiple studies have demonstrated that this DNA has many other functions (especially in epigenetic control). It is even more surprising as the following paragraph (Lines 58-76) contains the exact information to contradict the adoption of this particular view by the authors.
For some reason, authors decided to completely omit the growing piles of evidence in recent years that large parts of non-coding DNA play important role in the epigenetic control of gene and genome functioning. This leaves the overall discussion incomplete and the conclusions made at the end not well substantiated.
The authors' decision to start some subsections with question(s) seems to suppose that these questions will be answered in the respective section. Not surprisingly (as the field is still very much in development), most of these are left incompletely answered. All this creates unnecessary interruptions in the story flow, which makes the reading difficult. To my opinion, omitting the questions and just reporting and discussing current advancements would both make reading easier and avoid having questions put forth and then leaving them only partially answered.
The title of the first section ("Introduction (The concept of genetic redundancy ("selfish DNA") in eukaryotes") is needlessly long, which doesn't bring any specific benefits to it being so. It should be shortened to "Introduction" only.
The quality and assembling of the figures need to be much improved as the captions appear dissociated from them (i.e. Figure 1) and the graphics and legends are overlaying (Figure 2) in a way that makes reading them very difficult. In all cases, the figures and tables are improperly breaking the text flow of the associated paragraphs, which also needs to be corrected.
Very long sentences persist throughout the text (i.e. in Lines 319-352) that contain several ideas. This could be avoided if the separate ideas are separated in individual sentences, that would make the reading much easier.
Round 2
Reviewer 3 Report
If the authors are so confident in their way of writing the manuscript (supported by the overall evaluation of the other two reviewers), I will leave it to the Editor to decide if the authors' version is better suited for the journal than the modifications in the style and content I proposed.